# Image Reconstruction via Deep Image Prior Subspaces

**Riccardo Barbano**  *riccardo.barbano.19@ucl.ac.uk*
*Department of Computer Science*
*University College London*

**Javier Antorán**  *ja666@cam.ac.uk*
*Department of Engineering*
*University of Cambridge*

**Johannes Leuschner**  *jleuschn@uni-bremen.de*
*University of Bremen*

**José Miguel Hernández-Lobato**  *jmh233@cam.ac.uk*
*Department of Engineering*
*University of Cambridge*

**Bangti Jin**  *b.jin@cuhk.edu.hk*
*The Chinese University of Hong Kong*

**Željko Kereta**  *z.kereta@ucl.ac.uk*
*Department of Computer Science*
*University College London*

**Reviewed on OpenReview:** *https://openreview.net/forum?id=torWsEui9N*

## Abstract

Deep learning has been widely used for solving image reconstruction tasks but its deployability has been held back due to the shortage of high-quality paired training data. Unsupervised learning methods, e.g., deep image prior (DIP), naturally fill this gap, but bring a host of new issues: the susceptibility to overfitting due to a lack of robust early stopping strategies and unstable convergence. We present a novel approach to tackle these issues by restricting DIP optimisation to a sparse linear subspace of its parameters, employing a synergy of dimensionality reduction techniques and second order optimisation methods. The low-dimensionality of the subspace reduces DIP's tendency to fit noise and allows the use of stable second order optimisation methods, e.g., natural gradient descent or L-BFGS. Experiments across both image restoration and tomographic tasks of different geometry and ill-posedness show that second order optimisation within a low-dimensional subspace is favourable in terms of optimisation stability to reconstruction fidelity trade-off.

## 1 Introduction

Deep learning (DL) approaches have shown impressive results in a wide variety of linear inverse problems in imaging, e.g., denoising (Tian et al., 2020), super-resolution (Ledig et al., 2017; Ulyanov et al., 2020), magnetic resonance imaging (Zeng et al., 2021) and tomographic reconstruction (Wang et al., 2020). Mathematically, a linear inverse problem of recovering an unknown image $x \in \mathbb{R}^{d_x}$ from measurements $y \in \mathbb{R}^{d_y}$ is formulated as

$$y = Ax + \epsilon, \tag{1}$$

for an (ill-conditioned) matrix $A \in \mathbb{R}^{d_y \times d_x}$ and exogenous noise $\epsilon$.

However, conventional supervised DL approaches are not ideally suited for practical inverse problems. Large quantities of clean paired data, typically needed for training, are not available in many problem domains, e.g., tomographic reconstruction. Moreover, the ill-posedness (due to the forward operator $A$ and noise $\epsilon$) and high-dimensionality of the images $x$ pose significant challenges, and can be computationally very demanding. Whereas standard imaging tasks, e.g., denoising and deblurring, use high dimensional observations ($d_x \approx d_y$), tomographic imaging often requires reconstructing an image from observations that are of a much lower dimensionality compared to the sought-after images. For example, reconstructing a CT of a walnut may require reconstructing from observations that are only 3% of the size of the original image.

Unsupervised DL approaches do not require paired training data, and have received significant attention in the imaging community. Among these approaches, DIP (Ulyanov et al., 2018) has garnered the most traction. DIP parametrises the reconstructed image as the output of a convolutional neural network (CNN) with a fixed input. The reconstruction process learns low-frequency components before high-frequency ones (Chakrabarty & Maji, 2019; Shi et al., 2022), which can act as a form of regularisation.

Alas, the practicality of DIP is hindered by two key issues. Firstly, each DIP reconstruction requires training the entire network anew. This can take from several minutes up to a couple of hours for high-resolution images (Barbano et al., 2022b). Secondly, DIP requires careful early stopping (Wang et al., 2021) to avoid overfitting, which is often based on case-by-case heuristics. However, validation-based stopping criteria are often not viable in the unsupervised setting as they violate i.i.d. assumptions, see Wang et al. (2021).

This paper aims to address both of the existing issues inherent to DIP, and to illustrate the approach on image restoration and reconstruction. Building upon recent body of evidence which shows that neural network (NN) training often takes place in low-dimensional subspaces (Li et al., 2018; Frankle & Carbin, 2019; Li et al., 2023), we restrict DIP's optimisation to a sparse linear subspace of its parameters. This has a two-fold beneficial effect. First, subspace optimisation trades off some flexibility in capturing fine image structure details for robustness to overfitting. This is extraordinarily well suited in imaging problems belonging to inherently lower dimensional structures, but it is also shown to be competitive in restoring natural images. Moreover, it allows using stopping criteria based on the training loss, without sacrificing performance. Second, the low dimensionality induced by the subspace allows using second order optimisation methods (Amari, 2013; Martens & Grosse, 2015a). This greatly stabilises the reconstruction process, facilitating the use of a simple loss-based stopping criterion, and reduces the number of iterations to convergence.

Our contributions can be summarised as:

- We extract a principal subspace from DIP's parameter trajectory during a synthetic pre-training stage. To reduce the memory footprint of working with a non-axis-aligned subspace, we sparsify the extracted basis vectors using top-$k$ leverage scoring.

- We use second order methods: natural gradient descent (NGD) and limited-memory Broyden–Fletcher–Goldfarb–Shanno (L-BFGS), to optimise DIP's parameters in a low-dimensional subspace.

- We provide thorough experimental results across several linear inverse problems, comprising image restoration and tomographic tasks of different geometry and degree of ill-posedness, showing that subspace methods are favourable in terms of optimisation stability to reconstruction fidelity trade-off.

## 2 Related Work

The present work lies at the intersection of overfitting in unsupervised methods, subspace learning and second order optimisation. Below we discuss recent advances in the related fields.

**Avoiding overfitting in DIP:** Since the introduction of DIP (Ulyanov et al., 2018; 2020), stopping optimisation before overfitting to noise has been a necessity. The analysis in Chakrabarty & Maji (2019) and Shi et al. (2022) elucidates that U-Net is biased towards learning low-frequency components before high-frequency ones. The authors suggest a sharpness-based stopping criterion, which however requires a

modified architecture. Jo et al. (2021) propose a criterion based on the Stein's unbiased risk estimate (Eldar, 2008) for denoising, which however performs poorly for ill-posed settings (Metzler et al., 2018). Wang et al. (2021) propose a running image-variance estimate as a proxy for the reconstruction error. Our experiments find this method somewhat unreliable for sparse CT reconstruction. Ding et al. (2022) and Yaman et al. (2021) propose to split the observation vector into training and validation sub-vectors, and use the loss functional on the latter as a stopping criteria. Unfortunately, this violates the i.i.d. data assumption that underpins validation-based early stopping (Shalev-Shwartz & Ben-David, 2014, Theorem 11.2). Independently, Liu et al. (2019) and Baguer et al. (2020) add a TV regulariser to the DIP objective (2). This only partially alleviates the need for early stopping and has seen widespread adoption. To the best of our knowledge, the present work is the first to successfully avoid overfitting without significant performance degradation.

**Linear subspace estimation:**  Literature on low-rank matrix approximation is rich, with randomised SVD approaches being the most common (Halko et al., 2011; Martinsson & Tropp, 2020). However, in high-dimensions, even working with a small set of dense basis vectors can itself be prohibitively expensive. Matrix sketching methods (Drineas et al., 2012; Liberty, 2013) alleviate this through axis-aligned subspaces. To the best of our knowledge, our work is the first to combine these two method classes, producing non-axis-aligned but sparse approximations.

**Optimising neural networks in subspaces:**  Closely related to the present work is that of Li et al. (2018) and Wortsman et al. (2021), who find that networks can be trained in low-dimensional subspaces of the parameters without loss of performance, and that more complex tasks need larger subspaces. Similarly to our methodology, Li et al. (2023) identify subspaces from training trajectories and observe the resulting robustness to label noise. Results reported in Frankle & Carbin (2019) show that pruning a very large number of parameters in fully-connected and convolutional feed-forward networks yields trainable sub-networks that can achieve performance comparable to that of the original network. This principle has yielded speedups in network evaluation (Wen et al., 2016; Daxberger et al., 2021). Shwartz-Ziv et al. (2022) use a low-rank estimate of the curvature around an optimum of a pre-training task to regularise subsequent supervised learning.

**Second order optimisation for neural networks:**  Despite their adoption in traditional optimisation (Liu & Nocedal, 1989), second order methods are rarely used with neural networks due to the high cost of dealing with curvature matrices for high-dimensional functions. Martens & Sutskever (2012) use truncated conjugate-gradient to approximately solve against a network's Hessian. However, a limitation of the Hessian is that it is not guaranteed to be positive semi-definite (PSD). This is one motivation for NGD (Foresee & Hagan, 1997; Amari, 2013; Martens, 2020), that uses the FIM (guaranteed PSD). Commonly, the KFAC approximation (Martens & Grosse, 2015a) is used to reduce the costs of FIM storage and inversion. Also, common deep-learning optimisers, e.g., Adam (Kingma & Ba, 2015) or RMSprop (Hinton et al., 2014) may be interpreted as computing online diagonal approximations to the Hessian.

## 3 Deep Image Prior

DIP expresses the reconstructed image $x = f(x_0, \theta)$ in terms of the parameters $\theta \in \mathbb{R}^{d_\theta}$ of a CNN $f : \mathbb{R}^{d_x} \times \mathbb{R}^{d_\theta} \to \mathbb{R}^{d_x}$, and fixed input $x_0 \in \mathbb{R}^{d_x}$. The parameters $\theta$ are learnt by minimising the loss

$$\mathcal{L}(\theta) = \|Af(x_0, \theta) - y\|_2^2 + \lambda \text{TV}(f(x_0, \theta)), \tag{2}$$

composed of a data fidelity and total variation (TV), weighed by $\lambda > 0$. TV is the most popular regulariser for image reconstruction (Rudin et al., 1992; Chambolle et al., 2010). Its anisotropic version is given by

$$\text{TV}(x) = \sum_{i,j} |X_{i,j} - X_{i+1,j}| + \sum_{i,j} |X_{i,j} - X_{i,j+1}|, \tag{3}$$

where $X \in \mathbb{R}^{h \times w}$ is a vector $x \in \mathbb{R}^{d_x}$ reshaped into an $h \times w$ image, and $d_x = h \cdot w$. In this work, $f$ is a fully convolutional U-Net, see C.3 for more details. This implicitly regularises the reconstruction by preventing overfitting to noise, as long as the optimisation is stopped early enough.

DIP optimisation costs can be reduced by pre-training on synthetic data. E-DIP (Barbano et al., 2022b) first generates samples from a training data distribution $P$ of random ellipses, and then applies the forward model $A$ and adds white noise, following (1). The network input is set as the filtered back-projection (FBP) $x_0 = x^\dagger := A^\dagger y$, where $A^\dagger$ denotes the (approximate) pseudo-inverse of $A$. The pre-training loss is given by

$$\mathcal{L}_{\text{pre}}(\theta) = \mathbb{E}_{x,y \sim P}\|f(x^\dagger, \theta) - x\|_2^2, \tag{4}$$

where $P$ denotes the (empirical) joint distribution between the ground truth $x$ and the corresponding noisy data. The pre-trained network can then be fine-tuned on any new observation $y$ by optimising the objective (2) with FBP as the input. E-DIP decreases the DIP's training time, but can increase susceptibility to overfitting, making early stopping even more critical, cf. discussion in Section 5.

## 4 Methodology

We now describe our procedure for optimising DIP parameters in a subspace. We first describe how the E-DIP pre-training trajectory is used to extract a sparse basis for a low-dimensional subspace of the parameters. The objective is then reparametrised in terms of sparse basis coefficients. Finally, we describe how L-BFGS and NGD are used to update the sparsified subspace coefficients.

**Step 1 — Identifying the sparse subspace:**  First, we find a subspace of parameters that is low-dimensional and *easy to work with*, but contains a rich enough set of parameters to fit to the observation $y$. We leverage E-DIP pre-training trajectory to acquire basis vectors by stacking $d_{\text{pre}}$ parameter vectors, sampled at uniformly spaced checkpoints on the pre-training trajectory, into a matrix $\Theta^{\text{pre}} \in \mathbb{R}^{d_\theta \times d_{\text{pre}}}$. We then compute top-$d_{\text{sub}}$ SVD of $\Theta^{\text{pre}} \approx USV^\top$, and keep the left singular vectors $U \in \mathbb{R}^{d_\theta \times d_{\text{sub}}}$, where $d_{\text{sub}} \leq d_{\text{pre}}$ is the dimensionality of the chosen subspace. We then sparsify the orthonormal basis $U$ by computing leverage scores (Drineas et al., 2012), associated with each DIP parameter as

$$\ell_i = \sum_{k=1}^{d_{\text{sub}}} [U]_{ik}^2, \quad i = 1, \ldots, d_\theta.$$

We keep only the basis vector entries corresponding to $d_{\text{lev}} < d_\theta$ largest leverage scores. This can be achieved by applying a diagonal mask $M \in \{0,1\}^{d_\theta \times d_\theta}$ satisfying $[M]_{ii} = \mathbb{1}(i \in \arg \text{top-}d_{\text{lev}}\, \ell)$, where $\ell = [\ell_1, \ell_2, \ldots, \ell_{d_\theta}]$. The sparse basis $MU$ contains at most $d_{\text{lev}} \cdot d_{\text{sub}}$ non-zero entries.

Pre-training and sparse subspace selection are only performed once, and can be amortised across different reconstructions. We choose $d_{\text{pre}} \approx 10^3$, resulting in a large memory footprint of the matrix $\Theta^{\text{pre}}$, though this is stored in cpu memory. Alternatively, incremental SVD algorithms (Brand, 2002) can be used to further reduce the memory requirements (see Appendix A). Training DIP in the sparse subspace requires storing only the sparse basis vectors $MU$ in accelerator (gpu / tpu) memory. Thus, sparsification allows training in relatively large subspaces $d_{\text{sub}} > 10^3$ of large networks $d_\theta > 10^7$.

**Step 2 — Network reparametrisation:**  We write the objective $\mathcal{L}(\theta)$ in terms of sparse basis coefficients $c \in \mathbb{R}^{d_{\text{sub}}}$ as

$$\mathcal{L}_\gamma(c) := \mathcal{L}(\gamma(c)), \quad \text{with } \gamma(c) := \theta^{\text{pre}} + MUc. \tag{5}$$

This restricts the DIP parameters $\theta^{\text{pre}}$ (from pre-training) to change only along the sparse subspace $MU$. The coefficient vector $c$ is initialised as a sample from a uniform distribution on the unit sphere.

**Step 3 — Second order optimisation:**  The subspace reparametrised objective $\mathcal{L}_\gamma$ in (5) differs from traditional DL loss functions in that the method-dependent local curvature matrix of $\mathcal{L}_\gamma$ can be computed and stored accurately and efficiently, without resorting to restrictive approximation of its structure, such as the often used diagonal or KFAC approximations (Martens & Grosse, 2015b). These facts open the door to second order optimisation of $\mathcal{L}_\gamma$, which may converge faster than first order methods. However, repeatedly evaluating second order derivatives of neural networks has a prohibitive cost, further compounded by the rapid

change of the local curvature of the non-quadratic loss when traversing the parameter space. We mitigate this by performing online low-rank updates of a curvature estimate while only accessing loss Jacobians. In particular, we use L-BFGS (Liu & Nocedal, 1989) and stochastic NGD (Amari, 1998; Martens, 2020) in the experimental evaluation. The former estimates second directional derivatives by solving the secant equation. The latter keeps an exponentially moving average of stochastic estimates of the Fisher information matrix (FIM).

**NGD for DIP in a subspace:**     The exact FIM is given as

$$\mathbb{E}_{v \sim \mathcal{N}(Af(x^\dagger, \gamma(c)),\, I_{d_y})}[\nabla_c \mathcal{L}_\gamma(c)^\top \nabla_c \mathcal{L}_\gamma(c)], \tag{6}$$

where $\nabla_c \mathcal{L}_\gamma(c) = (Af(x^\dagger, \gamma(c)) - v)^\top AJ_f MU \in \mathbb{R}^{1 \times d_{\text{sub}}}$ is the Jacobian of the subspace loss $\mathcal{L}_\gamma$ at the current coefficients (with $I_{d_y} \in \mathbb{R}^{d_y \times d_y}$ being the identity matrix), and $J_f := \nabla_\theta f(x^\dagger, \theta)|_{\theta=\gamma(c)} \in \mathbb{R}^{d_x \times d_\theta}$ is the neural network Jacobian at $\theta = \gamma(c)$. Note that TV's contribution is omitted since the FIM is defined only for terms that depend on the observations and not for the regulariser (Ly et al., 2017). At step $t$ we estimate the FIM by the Monte-Carlo method as

$$\hat{F}_t = \frac{1}{n} \sum_{i=1}^n (z_i^\top AJ_f MU)^\top z_i^\top AJ_f MU, \quad \text{with } z_i \sim \mathcal{N}(0, I_{d_y}). \tag{7}$$

The moving average of the FIM is updated as

$$F_{t+1} = \beta F_t + (1 - \beta)\hat{F}_t \quad \text{with } \beta \in (0, 1). \tag{8}$$

Our implementation of NGD follows the approach of Martens & Grosse (2015a), with adaptive damping, and step size and momentum parameters chosen by locally minimising a quadratic model. See Appendix B for additional discussion.

Note that he methodology presented in this section can be extended beyond the DIP framework, in the spirit of test-time fine-tuning strategies of learned reconstruction methods (Darestani et al., 2022). Here we offer an alternative for adapting reconstruction algorithms at test-time, by introducing a robust optimisation framework.

# 5   Experiments

Our experiments cover a wide range of image restoration and tomographic reconstruction tasks. In Section 5.1, we conduct an ablation study on CartoonSet (Royer et al., 2017), examining the impact of subspace dimension $d_{\text{sub}}$, subspace sparsity level $d_{\text{lev}}$, degree of problem ill-posedness, and choice of the optimiser on reconstruction speed and fidelity. The acquired insights are then applied to challenging tomography tasks and image restoration. We compare the reconstruction fidelity, propensity to overfitting, and convergence stability relative to vanilla DIP and E-DIP on a real-measured high-resolution CT scan of a walnut, in Section 5.2, and on medically realistic high-resolution abdomen scans from the Mayo Clinic, in Section 5.3.

We simulate observations using (1) with dynamically scaled Gaussian noise given by

$$\epsilon \sim \mathcal{N}(0, \sigma^2 I_{d_y}) \quad \text{with } \sigma = p/d_y \sum_{i=1}^{d_y} |y_i|, \tag{9}$$

with the noise scaling parameter set to $p = 0.05$, unless noted otherwise. We conclude with denoising and deblurring on a standard RGB natural image dataset (Set5) in Section 5.4. For studies in Sections 5.2, 5.3 and 5.4, we use a standard fully convolutional U-Net architecture with either $\sim 3M$ (for CT reconstruction tasks) or $\sim 1.9M$ (for natural images) parameters, see architectures in Appendix C.3. For the ablative analysis in Section 5.1, we use a shallower architecture ($\sim .5M$) with only 64 channels and four scales, keeping the skip connections in lower layers. Following the literature, we train the vanilla DIP (Ulyanov et al., 2018) (labelled DIP) and E-DIP (Barbano et al., 2022b) with ADAM. We train subspace coefficients with Adam (Sub-DIP

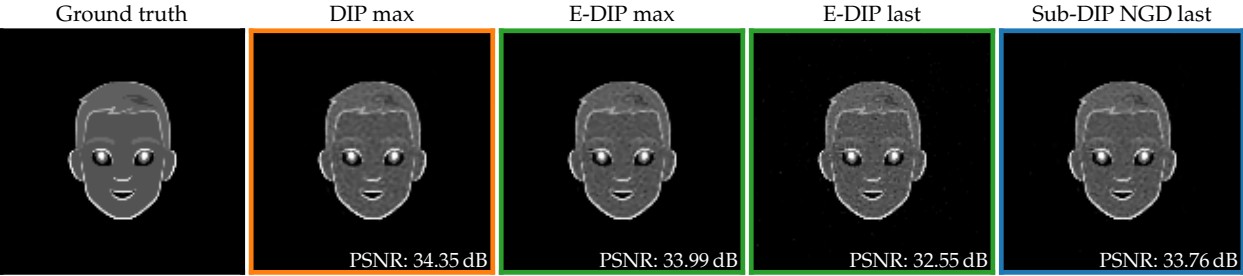

Figure 1: Reconstruction comparison for an example CartoonSet image from 45 angles. "max" indicates oracle (highest possible) PSNR, which we note is only available if a ground truth image exists. "last" denotes the final reconstruction.

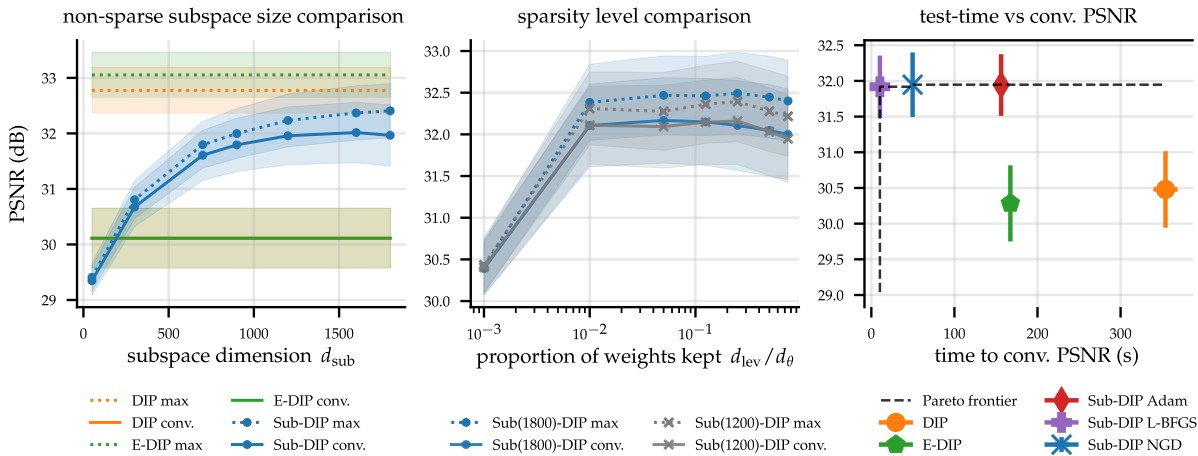

Figure 2: The influence of subspace dimension $d_{\text{sub}}$ (left), sparsity level $d_{\text{lev}}/d_\theta$ (middle) on PSNR, and the PSNR vs time Pareto-frontier (right) for CartoonSet. The dashed line defines a Pareto-frontier. "max" refers to the oracle stopping PSNR, while "conv." refers to the PSNR at the stopping point found by applying Algorithm 1, cf. Appendix A, to (2), with $\delta$=0.995 and patience of $\mathfrak{p}$=100. Left and middle plots use NGD. Results show mean and standard deviation over 25 reconstructed images from 95 angles. Runs are performed on A100 GPU. Note that the PSNR line for DIP conv., in the leftmost panel, is almost completely obscured by E-DIP conv. PSNR, since they have an almost identical PSNR value.

Adam), which serves as a baseline, L-BFGS (Sub-DIP L-BFGS) and NGD (Sub-DIP NGD). Image quality is assessed through peak signal-to-noise ratio (PSNR).

For tomographic tasks we use the same pre-training runs for E-DIP and all subspace methods: minimising (4) over $32k$ images of ellipses with random shape, location and intensity. Pre-training inputs are constructed from an FBP obtained with the same tomographic projection geometry as the dataset under consideration. Analogously, for image restoration tasks, we pre-train on ImageNet (Deng et al., 2009).

The method is built on top of the E-DIP library (`github.com/educating-dip`). The full implementation and data are available at `github.com/subspace-dip`.

## 5.1 Ablative analysis on CartoonSet (Royer et al., 2017)

We investigate Sub-DIP's sensitivity to subspace dimension $d_{\text{sub}}$, sparsity level $d_{\text{lev}}$, and ill-posedness on 25 images of size $(128\,\text{px})^2$ from CartoonSet (Royer et al., 2017, `google.github.io/cartoonset`). Example reconstructions are shown in Fig. 1. We simulate a parallel-beam geometry with 183 detectors and 45, 95 or 285 angles, corresponding to, respectively, a very sparse-view ($d_y$=8235), a moderately sparse-view ($d_y$=17385), and a fully sampled setting ($d_y$=52155). We construct subspaces by sampling $d_{\text{pre}}$=2$k$ parameters

at uniformly spaced checkpoints during 100 pre-training epochs on ellipses as in Barbano et al. (2022b). We measure the degree of overfitting by comparing the highest PSNR obtained throughout optimisation (max) with that given by Algorithm 1, applied to the training loss (2) with $\delta$=0.995 and $\mathfrak{p}$=100 steps (conv.).

In Appendix A we report results of additional ablation experiments on the extracted subspaces, examining their adaptability to changes in the forward operator, and comparing them to random subspace bases.

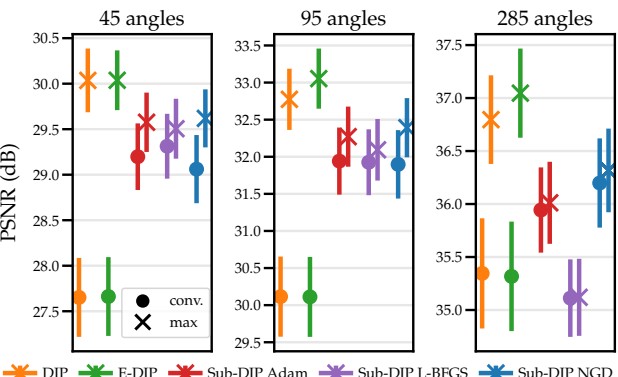

**Subspace dimension:** Fig. 2 (left) explores the trade-off between subspace dimension $d_{\text{sub}}$ and reconstruction quality. We use 95 angles, subspace methods with no sparsity ($d_{\text{lev}}=d_\theta$) and the NGD optimiser. Both standard DIP and E-DIP overfit, showing a $\approx 3$ dB gap between max and conv. PSNR values, while subspace methods exhibit only a $\approx 0.5$ dB gap. Both max and conv. PSNR present a monotonically increasing trend with subspace dimension, while the gap at $d_{\text{sub}} > 1k$ stays roughly constant at

Figure 3: PSNR mean and standard deviation over 50 CartoonSet images from 45, 95, and 285 angles.

$\sim$0.25 dB. Thus, these subspace dimensions are too small for significant overfitting to occur. In spite of this, $d_{\text{sub}}$=100 is enough to obtain better conv. PSNR than DIP and E-DIP.

**Subspace sparsity level:** Fig. 2 (middle) shows that for $d_{\text{lev}}/d_\theta > 0.01$, the reconstruction fidelity is largely insensitive to the sparsity level. This is true for both $d_{\text{sub}}$=1200 and $d_{\text{sub}}$=1800. This effect is also somewhat independent of the subspace dimensionality $d_{\text{sub}}$. Hence, *sparse subspaces should be constructed by choosing a large $d_{\text{sub}}$ and then sparsifying its basis vectors to ensure computational tractability.*

**Problem ill-posedness:** We report the performance with varying numbers of view angles in Fig. 3, with non-sparse subspaces of dimension $d_{\text{sub}}$=1800. The smaller is the number of view angles, the more ill-posed the reconstruction problem becomes. Subspace methods present a smaller gap between max and conv. PSNR ($\sim 0.25$ dB) than DIP and E-DIP ($\sim 2.5$ dB). Subspace methods also present better fidelity at convergence for all the studied number of view angles. Notably, the improvement is larger for more ill-posed settings ($\sim 2.5$ dB at 45 and 95 vs $\approx 1$ dB at 285 angles), despite the worsening performances for all methods in sparser settings. This is expected as there is a reduced risk of overfitting in data-rich regimes and more flexible models can do better. E-DIP's max reconstruction fidelity is consistently above that of other methods by at least 0.5 dB. This may be attributed to full-parameter

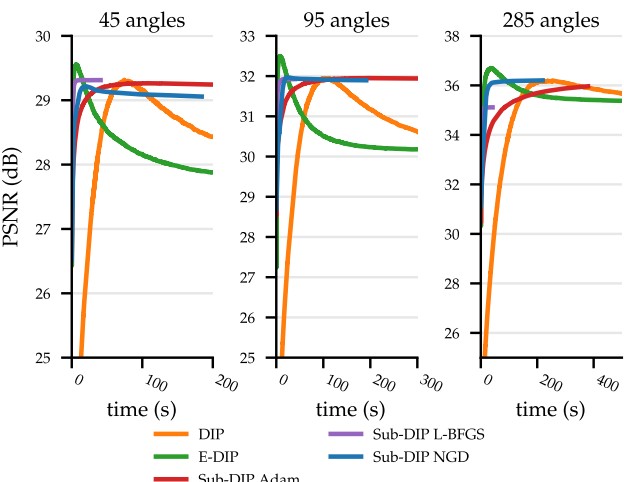

Figure 4: Average PSNR trajectories for 45, 95, and 285 angles over 50 reconstructions.

flexibility with benign inductive biases from pre-training. However, obtaining max PSNR performance requires oracle stopping and is not achievable in practice. In Fig. 4, we show PSNR trajectories of the studied methods (DIP, E-DIP, Sub-DIP Adam, Sub-DIP LBFGS, and Sub-DIP NGD) for 45, 95, and 285 angles, averaged over 50 reconstructed images. This figure provides additional evidence supporting previous observations. While DIP and E-DIP tend to overfit to noise once they reach their maximum PSNR values, subspace methods consistently maintain stable reconstruction performance with no noticeable performance degradation, due to the inherent regularising effect obtained by restricting to a subspace.

**First vs second order optimisation:** We compare optimisers' conv. PSNR vs their time to convergence. Fig. 2 (right) shows that Sub-DIP L-BFGS and NGD converge in less than 50 seconds. These methods are optimal along the known Pareto-frontier, reaching $\sim 1.5$ dB higher reconstruction fidelity than DIP and E-DIP. LBFGS converges in only $\sim 20$ seconds. Sub-DIP Adam retains protection against overfitting but converges at a rate similar to non-subspace first order methods (in $\sim 180$ seconds). These trends hold across studied degrees of ill-posedness; see Appendix A.1.

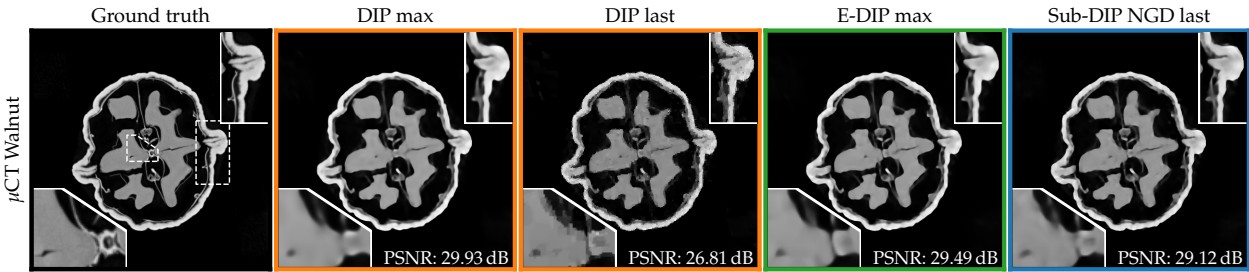

Figure 5: Reconstructions of the Walnut using 60 angles. Sub-DIP reconstructions do not capture any noise, but present slightly increased ringing around very thin structures. "max" indicates oracle (highest possible) PSNR. "last" denotes the final reconstruction.

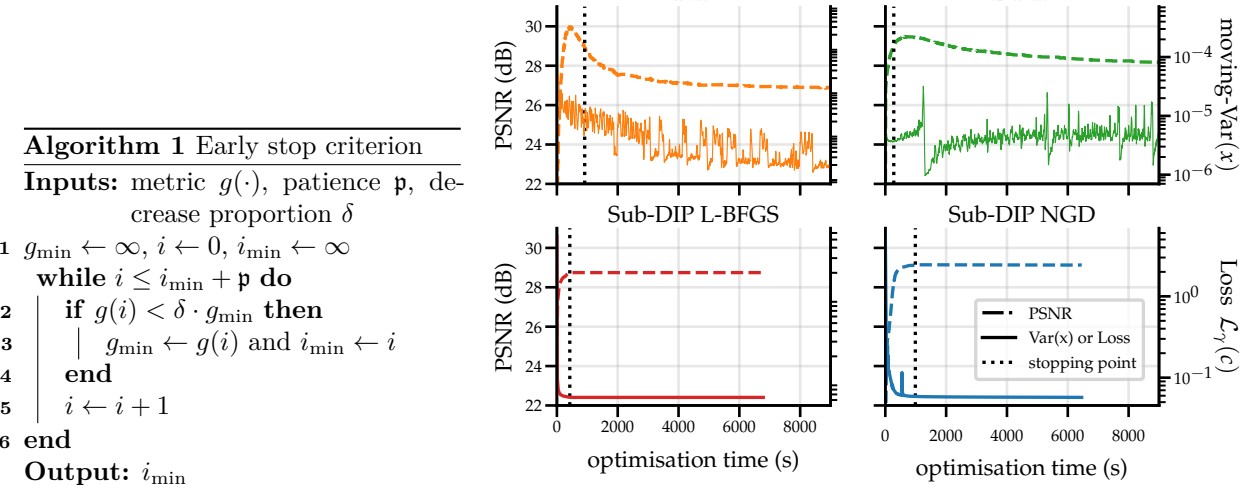

**Algorithm 1** Early stop criterion

**Inputs:** metric $g(\cdot)$, patience $\mathfrak{p}$, decrease proportion $\delta$

1  $g_{\min} \leftarrow \infty$, $i \leftarrow 0$, $i_{\min} \leftarrow \infty$

    **while** $i \leq i_{\min} + \mathfrak{p}$ **do**

2      **if** $g(i) < \delta \cdot g_{\min}$ **then**

3          $g_{\min} \leftarrow g(i)$ and $i_{\min} \leftarrow i$

4      **end**

5      $i \leftarrow i + 1$

6  **end**

    **Output:** $i_{\min}$

Figure 6: The evolution of PSNR and stopping metrics (variance-based for DIP and E-DIP, and loss-based for Sub-DIP NGD and L-BFGS) vs optimisation time on Walnut with one seed.

## 5.2 µCT Walnut (Der Sarkissian et al., 2019)

In this section we study the methods on a real-measured high-resolution µCT problem. We reconstruct a $(501\,\mathrm{px})^2$ slice from a very sparse, real-measured cone-beam scan of a walnut, using 60 angles and 128 detector pixels ($d_y = 7680$). We compare reconstructions against ground-truth (Der Sarkissian et al., 2019), which uses classical reconstruction from full data acquired at 3 source positions with 1200 angles and 768 detectors. This task involves fitting both broad-stroke and fine-grained image details (see Fig. 5), making it a good proxy for µCT of industrial context. We pre-train the 3M parameter U-Net for 20 epochs and save $d_{\mathrm{pre}}{=}5k$ parameter checkpoints. Following Section 5.1, we construct a $d_{\mathrm{sub}}{=}4k$ dimensional subspace and sparsify it down to $^{d_{\mathrm{lev}}}/_{d_\theta}{=}0.5$ of the parameters.

Fig. 7 (left) shows the optimisation curves for all optimisers, averaged across 3 seeds. Qualitatively, the results are similar to the cartoon data. Second order subspace methods converge to their highest reconstruction fidelity within the first 500 seconds and do not overfit. Vanilla DIP and E-DIP also converge quickly but

suffer from overfitting leading to ~3 dB and ~ 1.8 dB of performance degradation, respectively. Sub-DIP Adam takes over 3000 seconds to converge and does not overfit.

**Stopping criterion challenges:** Capturing the oracle performance of DIP and E-DIP would require a robust stopping criterion. Note that we cannot base a stopping criterion on (2). We instead turn to the method from Wang et al. (2021), which minimises a rolling estimate of the reconstruction variance across optimisation steps, a proxy for the squared error. Following Wang et al. (2021), we compute variances with a 100 step window and apply Algorithm 1 with a patience of $\mathfrak{p}$=1000 steps and $\delta$=1. Figure 6 shows this metric to be very noisy when applied to DIP and E-DIP. This breaks the smoothness assumption implicit in the stopping criterion (Algorithm 1), leading to stopping more than 1 dB before/after reaching the max PSNR. This phenomenon is attributed to the severe ill-posedness of the tasks causing variance across a large subspace of reconstructions that fit our observations well. For tomographic reconstruction, the variance curve becomes more non-convex, and its minimum does not correspond to the optimal PSNR. Since subspace methods do not overfit and the loss is smooth, we can use it as our stopping metric ($\delta$=0.995, $\mathfrak{p}$ = 100).

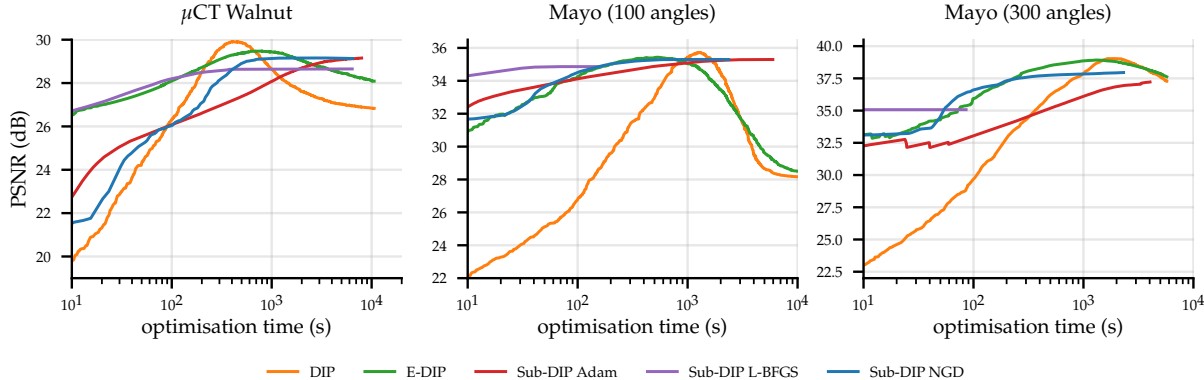

Figure 7: Optimisation curves for the $\mu$CT Walnut data (left), and Mayo using 100 angles (middle) and Mayo using 300 angles (right) averaged over 10 images.

### 5.3 Mayo Clinic dataset (Moen et al., 2021)

To investigate a medical setting, we use 10 clinical CT images of the human abdomen released by Mayo Clinic (Moen et al., 2021) and study the behaviour of subspace optimisation. We reconstruct from a simulated fan-beam projection, with 300 angles and white noise with the noise scaling parameter $p = 0.025$, cf. (9). As a more ill-posed reference setting, we use 100 angles (comparable sparsity to the Walnut setting) and Gaussian noise with $p$=0.05. For both tasks, we use the 3M parameter U-Net, as in Section 5.2, pre-trained on $32k$ ellipse samples. For the sparse setting (100 angles), we use a $d_{\mathrm{sub}} = 4k$ dimensional subspace constructed from $d_{\mathrm{pre}}$=5$k$ checkpoints, but with sparsity ratio $d_{\mathrm{lev}}/d_\theta$=0.25. For the more data-rich setting (300 angles), we use $d_{\mathrm{sub}} = 8k$, sampled from $d_{\mathrm{pre}}$=10$k$ checkpoints, and similarly, we sparsify it down to $d_{\mathrm{lev}}/d_\theta$=0.25.

In the 300 angle setting, Sub-DIP NGD reaches 37 dB within 200 seconds. PSNR the continutes to increas, albeit slowly, without performance saturation, cf. Fig. 7 (right). In contrast, for the sparser Walnut and Mayo data, Sub-DIP NGD maintains a steep PSNR increase until reaching max PSNR. Interestingly, L-BFGS does not perform well in the 300 angle setting, obtaining $< 36$ dB PSNR. This might be due to L-BFGS's tendency to stop the iterations too early in high dimensions.

Note that the observed time efficiency of the Sub-DIP reported in Fig. 2, is not preserved in high-dimensional CT tasks; see Fig. 7. This behaviour is expected as the efficiency of second order optimisation methods is affected both by the high-dimensionality of the measurements—resulting in costlier forward operator evaluations—and more importantly, by the depth of the U-Net—resulting in costlier vector-Jacobian and Jacobian-vector products. As discussed, second order methods converge in fewer iterations (see e.g., Fig. 19 in Appendix A) but require a higher per-iteration cost.

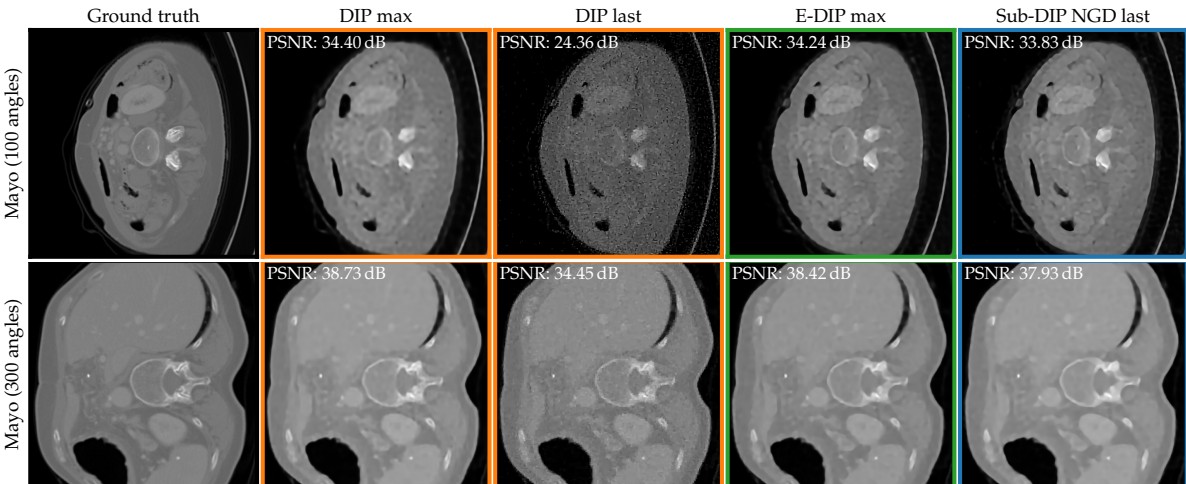

Figure 8: Example reconstructions on the Mayo dataset in 100 (top) and 300 (bottom) angle settings.

However, it is worth noting that when working with a limited runtime budget, Sub-DIP (L-BFGS and NGD) methods may take more time to reach their maximum PSNR, but they consistently deliver the highest image quality within a 100-second runtime, as illustrated in Fig. 7. At the same time, Sub-DIP methods do not suffer performance degradation due to overfitting to noise. A comparison in terms of the number of iterations, instead of optimisation time, can be found in the rightmost panel in Figs. 19, 20 and 21 in the Appendix.

Fig. 8 shows example reconstructions using 100 and 300 angles. If stopped within a narrow max PSNR window, DIP and E-DIP can deliver reconstructions that better capture high-frequency details than Sub-DIP methods as expected. However, while DIP and E-DIP reconstructions become noisy once they start to overfit, Sub-DIP methods do not exhibit any noise. We deem the *increased robustness vs reduced flexibility* trade-off provided by the Sub-DIP to be favourable, even in the well-determined setting.

In Appendix A.3 we investigate the effect of the dataset used to extract the subspace, in the 300 angle case. Namely, we compare the task-agnostic dataset of random ellipses with a task-specific Mayo dataset. As expected, the results show that using a task-specific dataset, matching the target image manifold, improves reconstruction performance. However, the generality of the approach is compromised, since a suitable dataset of images specific enough for a task and modality at hand might not always be available and the method starts to resemble supervised learning. In the data-poor settings, using a synthetic dataset provides a good performance compromise.

### 5.4 Image restoration of Set5

We conduct denoising and deblurring on five widely used RGB natural images ("baboon", "jet F16", "house", "Lena", "peppers") of size $(256\,\text{px})^2$. The pre-training is done on ImageNet (Deng et al., 2009), a dataset of natural images, which we use to extract the basis for the subspace methods.

For denoising we consider four noise settings with the noise scaling parameter $p \in \{0.10, 0.15, 0.25, 0.5\}$, cf. (9), We extract a single subspace for all the noise levels. To this end, during the pre-training stage, we construct a dataset by adding noise to each training datum, with a randomly selected noise scaling parameter $p \sim \text{Uni}(0.05, 0.5)$. Then a $d_{\text{sub}} = 8k$ subspace with sparsity level $d_{\text{lev}}/d_\theta = 0.25$ is extracted from $d_{\text{pre}} = 10k$ samples. In the high noise case ($p = 0.5$), we sub-extract to a smaller $d_{\text{sub}} = 1k$ subspace. This sub-extraction is necessary to minimise overfitting in high noise level scenarios. Further comments on this are at the bottom of the page. For deblurring we consider two settings, using a Gaussian kernel with std of $\kappa \in \{0.8, 1.6\}$ pixels and $p = 0.05$ Gaussian noise. We then follow an analogous procedure; adding $p = 0.05$ Gaussian noise and applying $\kappa \sim \text{Uni}(0.4, 2)$ blur to each training datum, and then constructing a single subspace.

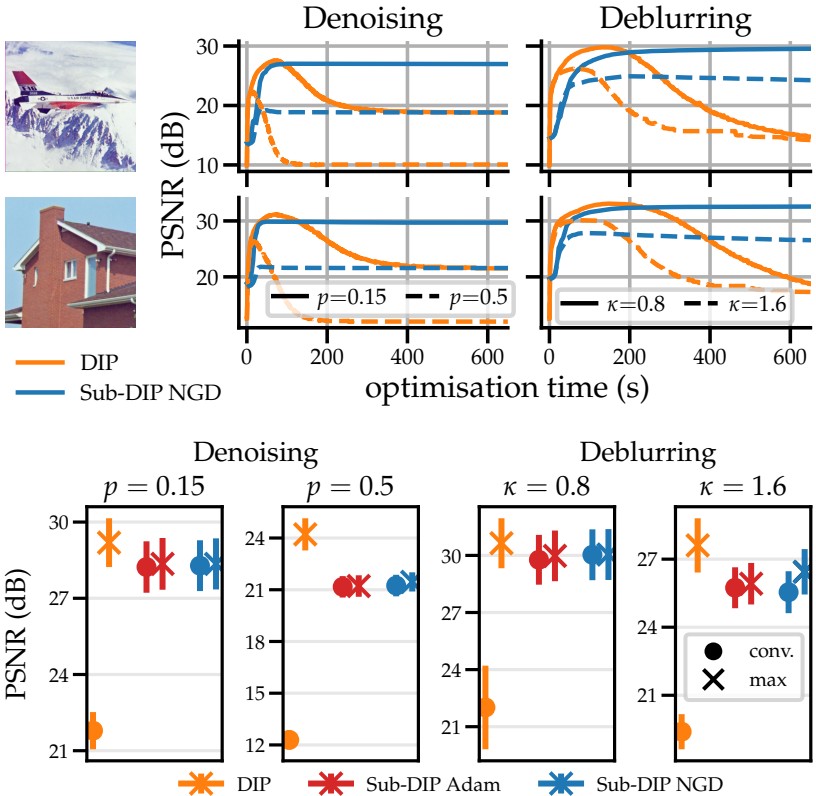

Figure 9: Denoising ($p = \{0.15, 0.5\}$) and deblurring ($\kappa = \{0.8, 1.6\}$) on the "jet F16" and "house" images. On the top we show PSNR trajectories for each of the tasks, comparing DIP and Sub-DIP NGD, and on the bottom we show mean and std of max and conv. PSNR over the studied 5 images. Results for the other two noise levels are reported in Appendix A.

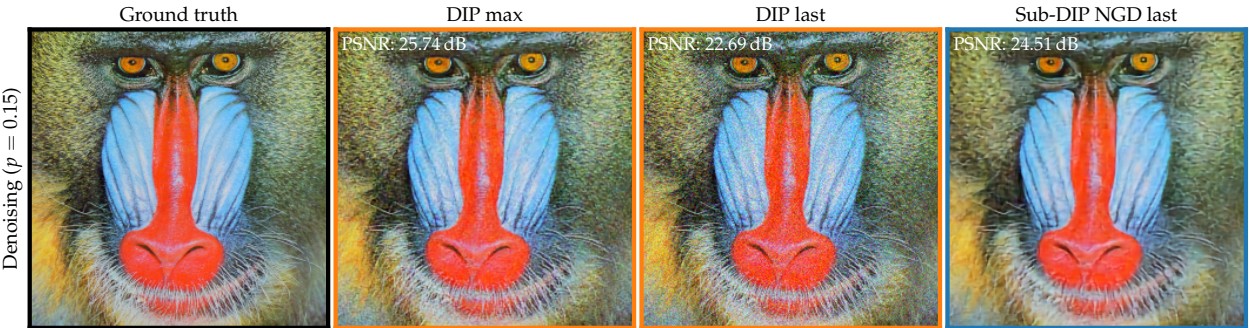

Figure 10: Restoration of the noisy (with noise scaling parameter $p = 0.15$) "baboon" image.

As common practice when deploying DIP on restoration tasks, we do not include the TV ($\lambda = 0$) in (2) for neither denoising nor deblurring. Instead, *the regularising property of the reconstruction stems exclusively from restricting the DIP optimisation to a low-dimensional subspace of its parameters.*

The top row in Fig. 9 shows PSNR trajectories for denoising and deblurring on "jet F16" and "house" images. These images contain large regions of continuous colour intensity with clearly defined edges. Hence, we expect subspace methods to perform well. This is confirmed by the results: Sub-DIP NGD and DIP have a comparable max PSNR. The bottom row in Fig. 9 shows the mean and std of max and conv. PSNR over 5 studied natural images for low and high noise and blur. The conv. reconstructions are computed by Algorithm 1, applied to the training loss (2) with $\delta$=0.995 and $\mathfrak{p}$=100 steps. We notice that DIP mildly

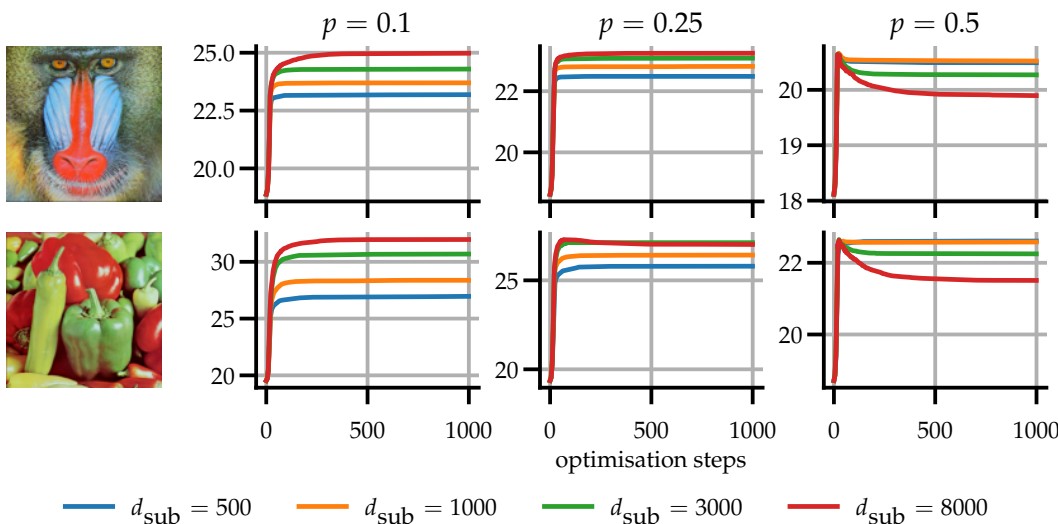

Figure 11: Investigation of the regularising effect of the dimensionality of the chosen subspace on Set5. We report PSNR trajectories for two specific images, the "baboon" and the "peppers" using Sub-DIP NGD. Our analysis encompasses three distinct noise levels and four dimensions of the subspace.

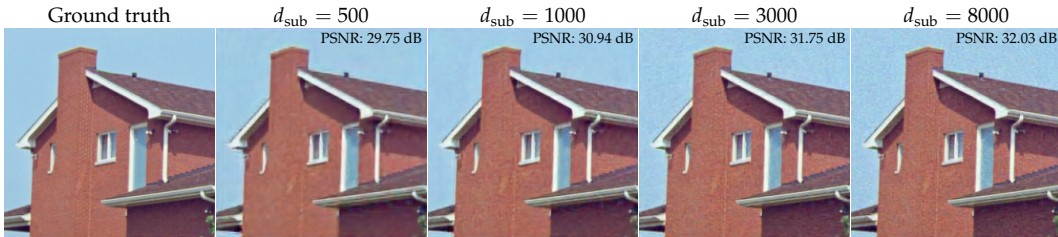

Figure 12: Restoration of the noisy (with noise scaling parameter $p = 0.1$) "house" image in Set5 using different subspace dimensions.

outperforms Sub-DIP in terms of time to max PSNR. However, DIP is also significantly more prone to overfit to noise and exhibits a sharp decline in image quality after reaching max PSNR. Thus, selecting a good stopping criterion is not only critical but also challenging.

Fig. 10 shows the denoising results for the "baboon" image with a moderate noise level ($p = 0.15$). The "baboon" image contains fine-grained details and sharp transitions throughout the image. Hence, we expect a somewhat comparatively worse performance of subspace methods on this task, as higher frequency information can be lost when using low-dimensional approaches. The results confirm this intuition: the max PSNR for the "baboon" with E-DIP is at least 1 dB higher than that for subspace methods. Moreover, while standard DIP experiences a sharp decline after achieving max PSNR, subspace methods retain the performance.

To gain deeper insight into the interplay between the dimension of the subspace and the noise level corrupting the data, we sweep over $d_{\text{sub}} \in \{500, 1000, 3000, 8000\}$ and $p \in \{0.1, 0.15, 0.25, 0.5\}$. While we maintain the ratio $d_{\text{lev}}/d_\theta$ at 1 for subspaces up to $3k$, for computational feasibility we reduce it to $0.25$ for the $8k$-dimensional case. As shown in Fig. 11, and in Fig. 23 and Fig. 22 in Appendix A.4, in high-noise scenarios (i.e., $p = 0.5$), using a larger subspace leads to overfitting. Conversely, in low-noise settings, a lower-dimensional subspace limits the network's capacity to fit higher frequencies and adapt, resulting in overly smooth reconstructions, cf. Fig. 12. We can thus conclude that *noisier problems require more regularisation, thus a lower-dimensional subspace may be beneficial.*

# 6 Conclusion

In this work, we develop a novel approach that constrains DIP optimisation to a sparse principal low-dimensional subspace, extracted from pre-training trajectories. This greatly mitigates, if not completely eliminates, overfitting. Our approach may be understood from the perspective of the bias-variance trade-off. At initialisation, the vanilla DIP presents a useful bias towards learning low-frequency image components; but the over-parameterisation of the neural network leads to overfitting. Pre-training only partially removes DIP's low pass bias, allowing the E-DIP to often fit images quickly. This comes at the cost of increased variance. Our approach seeks to efficiently navigate the bias-variance trade-off. Constraining the optimisation to a low-dimensional subspace greatly reduces the variance. By extracting the subspace from principal directions of pre-training trajectories, and through the use of leverage scoring, we limit the bias introduced into the model. Furthermore, optimising in lower dimensional subspaces allows using fast and stable second order optimisation methods. In future work, alternative approaches for the identification of the subspace and their implicit regularising properties will be investigated.

In our experiments on several image restoration and tomographic tasks, subspace DIP methods deliver reconstructions on par with DIP's max performance, and result in better reconstruction quality than the overfit DIP reconstructions.

## Acknowledgements

R.B. acknowledges the support from the i4health PhD studentship (UK EPSRC EP/S021930/1), and from The Alan Turing Institute (UK EPSRC EP/N510129/1). J.A. acknowledges the support from Microsoft Research, through its PhD Scholarship Programme, and from the EPSRC. J.L. was funded by the German Research Foundation (DFG; GRK 2224/1) and by the Federal Ministry of Education and Research via the DELETO project (BMBF, project number 05M20LBB). J.M.H.-L. acknowledges support from a Turing AI Fellowship EP/V023756/1 and an EPSRC Prosperity Partnership EP/T005386/1. The work of B.J. is partially supported by UK EPSRC grant EP/V026259/1, Hong Kong RGC General Research Fund (Project 14306423), and a start-up fund from The Chinese University of Hong Kong. Z.K. acknowledges the support from the UK EPSRC grant EP/X010740/1. This work has been performed using resources provided by the Cambridge Tier-2 system operated by the University of Cambridge Research Computing Service (http://www.hpc.cam.ac.uk) funded by EPSRC Tier-2 capital grant EP/T022159/1.

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
