# OpenReview forum: "Image Reconstruction via Deep Image Prior Subspaces"
_TMLR — Accepted by TMLR_

### Review · Reviewer_HM4E · 2023-10-13

**Summary Of Contributions:**

This work considers solving imaging inverse problems in low-data regimes. The Deep Image Prior (DIP) has shown to be a powerful implicit prior without requiring paired training data, but suffers from long runtimes and the need for early stopping to avoid overfitting. This work develops a method to mitigate both issues by performing inversion over an extracted low-dimensional principal subspace of parameters (found by pertaining on synthetic data). The low-dimensionality and sparsity of the subspace aid in reducing overfitting. Once this subspace is constructed, one can then leverage second-order optimization methods such as L-BFGS or natural gradient descent to improve the DIP’s runtime. The method is demonstrated on several datasets and inverse problems, including denoting, deblurring, and tomographic reconstruction.

**Audience:**

Yes

**Claims And Evidence:**

Yes

**Requested Changes:**

Answers to the following questions and comments would, in my opinion, strengthen the work.
- Did the authors analyze the relationship between the amount of noise/undersampling was present in the measurements versus the size and sparsity of the chosen subspace? If not, do the authors have a sense of what would be ideal choices for these parameters as a function of the amount of noise is present? I would suspect that for noisier problems, more regularization may be necessary, hence choosing a lower-dimensional subspace with a sparser basis may be beneficial.
- What would be some of the challenges in bringing this approach to nonlinear settings? In principle, I believe this could also work for problems such as phase retrieval or one-bit compressed sensing, but perhaps a discussion of these challenges or a simple example demonstrating the approach in a nonlinear setting could be interesting to include.
- How sensitive is the approach to distribution shifts in the forward model and images to be recovered, e.g., if one pre-trained with fewer/more angles than one sees at inversion time or one used synthetic images to recover natural images?

**Strengths And Weaknesses:**

## Strengths:
- The method is intuitive and well-motivated. To the reviewer’s knowledge, while pre-training the DIP has been proposed in the past, re-parameterizing optimization over low-dimensional subspaces and allowing for more efficient second-order optimization is new.
- There are clear speedups in runtime and overfitting is reduced.
- The experiments are quite extensive, and the authors do a good job of ablating various aspects of their method.
- The results would be of interest to the DL + inverse problems community.

## Weaknesses:
- While early stopping is mitigated, new hyper parameters are introduced that have a big influence on performance (e.g., size of the subspace and its sparsity pattern). Authors do, however, prescribe recommendations for how to set these parameters, but tuning is still required.
- The approach is only demonstrated for linear inverse problems.

---

> ### Author Response · Authors · 2023-12-20
> **Response to reviewer HM4E**
>
> We thank the reviewer for their time reading our paper. We go on to address their questions and concerns.
>
> ---
>
> **Addressing the interplay of noise and subspace selection:**
> >Did the authors analyze the relationship between the amount of noise/undersampling was present in the measurements versus the size and sparsity of the chosen subspace?
>
> As the reviewer suggested, **choosing a lower-dimensional subspace** when dealing with **high-noise setting enhanced regularisation**, making the reconstruction more **robust to overfitting**.
>
> To investigate this behaviour better, **we conduct an additional experiment using natural images**, looking at the interplay between subspace dimensionality and noise level.
>
> The reasons for using natural images and not the CartoonSet dataset are two-fold:
>
> 1. The subspace for natural images is extracted from training with images that have been distorted with randomised noise levels, i.e.,  $p\sim\rm{Uni}(0.05,0.5)$, **making the subspace extraction independent to varying noise levels**. This contrasts with the CartoonSet experiments, where the amount of noise influences the subspaces' extraction. For the CartoonSet, we train to post-process the filtered back-projection of noisy data, which is inherently dependent on the noise in simulated measurements.
>
> 2. The CartoonSet experiments incorporate Total Variation (TV) regularisation in the objective, a standard practice in CT reconstruction. However, the TV is absent in the objective used for the reconstruction of the natural images.
>
> **Our findings**, now reported at the end of **Section 5.4** and shown in **Figure 11** on page 12 as well as in **Appendix A.4**, align with the reviewer's intuition: in high-noise scenarios (e.g., $p=0.5$), **using a larger subspace leads to overfitting** (e.g., $d_{\textrm{sub}} = 3k $ or $8k$). Conversely, in low-noise settings, a lower-dimensional subspace limits the network's ability to fit higher frequencies and adapt, resulting in overly smooth reconstructions.
>
> ---
>
> **Subspace sensitivity to distributional shift in the forward operator:** We agree with the reviewer and recognise that this is an important question for many reconstruction methods for inverse problems.
> >How sensitive is the approach to distribution shifts in the forward model and images to be recovered [...]
>
> We conduct **additional experiments to explore how subspace sensitivity reacts to shifts in the forward operator**. We consider two sets of experiments on the CartoonSet. In the first set of experiments we extract a subspace from 45 angles and then apply reconstruction to settings of 95 and 285 angles. In the second set of experiments we extract a subspace from 285 angles and apply it to settings with 45 and 95 angles. Our findings, now reported in Appendix A.1, reveal that transferring the subspace extracted at 285 angles for testing on 45 and 95 angles yields max PSNR values nearly identical to those obtained when extracting and testing on 45 or 95 angles separately.
>
> Our experiments suggest that the subspace extracted is **robust to changes in the forward operator**.
>
> ---
>
> **What would be the challenges in bringing this approach to nonlinear settings:**
> >What would be some of the challenges in bringing this approach to nonlinear settings?
>
> Our work only deals with linear inverse problems, however here it follows **a discussion highlighting the challenges in scaling this method to non linear problems**.
>
> The primary challenge for general non-linear inverse problems would be in the computational issues arising from the Fisher Information Matrix (FIM). For a non-linear problem $\mathcal{F}(x)=y$, we have by [Bandeira et al.](https://doi.org/10.1016/j.acha.2013.10.002)
>
> $F(x)_{i,j}\propto\sum_n \frac{\partial}{\partial x_i}(\mathcal{F}(x))_n\frac{\partial}{\partial x_j}(\mathcal{F}(x))_n,$
>
> where $x$ is the output of the network. Whereas in the case of a linear inverse problem the above gradients are again a linear function of the input $x$, this is not the case for non-linear problems. For some specific types of problems the above expression could be simplified, for example, FIM for phase retrieval in the additive white Gaussian noise model is given by $F(x) \propto \Psi(x)\Psi(x)^\ast$, where $\Psi(x)$ is an $M\times N$ matrix with k-th column equal to $\langle x,\phi_k\rangle \phi_k$, where $\phi_k$ is the element of the corresponding measurement frame. However, for other nonlinear modalities, such as electrical impedance tomography, this does not admit a simple form.
>
> Furthermore, **the effectiveness of subspace extraction depends on our ability to easily simulate measured data** from ground-truth images. In many non-linear settings, simulating data often presents **a significant computational challenge**, e.g., in electrical impedance tomography the forward operator results in solving a PDE, which can be time-consuming, and computing the adjoint of the Frechet derivative is also computationally costly.

---

### Review · Reviewer_aq9Y · 2023-10-18

**Summary Of Contributions:**

Deep image prior (DIP) for image reconstruction/restoration is often doomed to produce poor results due to its tendency to eventually overfit the measurements when run till convergence. This work proposes a technique to regularize DIP by representing the parameters of the network in a lower dimensional space. The work claims to achieve improved empirical performance in terms of traditional measures of image quality, such as PSNR, as compared to conventional DIP on. variety of datasets.

**Audience:**

Yes

**Broader Impact Concerns:**

No broader impact concerns.

**Claims And Evidence:**

Yes

**Requested Changes:**

I think it is important that the authors add further intuition for why the specific scheme of dimensionality reduction is good. Additionally, I think it would be important to present this paper as a preliminary investigation of a new regularization scheme for DIP, and tone down statements that indicate that this paper brings DIP closer to real-life applications, since based on the results presented, it is hard to conclude this. It is also essential to mention the limitations of the evaluation methods used in the context of real-life applications.

Using a wider variety of evaluation metrics as well as task-based evaluation metrics will definitely strengthen the paper. Also,
it seems that the basis vectors used for the dimensionality reduction were extracted from a pretraining scheme based on an ellipse dataset. What sort of biases does this dataset introduce? Does this favor better reconstruction of piecewise smooth images rather than images with fine details? A discussion of this will also strengthen the paper.

**Strengths And Weaknesses:**

Strengths:
1. The approach is interesting and novel. It is definitely conceivable that reducing the dimensionality of the network parameters in some way will provide additional regularization.
2. The ablation studies presented are fairly comprehensive.

Weaknesses:
1. I think the paper does not provide enough intuition as to why the particular scheme of reducing the dimensionality of the network parameter vector works. Thus, the choices feel a bit arbitrary.
2. I think the evaluation methodology used is fairly limited and has several drawbacks. For example, in the case of medical imaging, improved performance of relevant downstream tasks (eg. detection tasks relevant for diagnosis) are far more important than PSNR. For applications where perceptual quality is the most important (eg. upsampling in computational photography), PSNR is still not always a good metric since it favors images with oversmoothed features. Thus, the authors should consider using a a better variety of metrics that are more suited to various applications considered in the paper.

---

> ### Author Response · Authors · 2023-12-20
> **Response to reviewer aq9Y – Part I**
>
> We thank the reviewer for their time reading our paper and for their comments and suggestions. We go on to address their questions and concerns.
>
> ---
>
> **1. The paper does not provide enough intuition as to why the particular scheme of reducing the dimensionality of the network parameter vector works.**
>
> > I think it is important that the authors add further intuition for why the specific scheme of dimensionality reduction is good.
>
> Thanks for highlighting this! **We provide some additional intuition below, which we have added to the introduction.**
>
> Ultimately, our method is heuristic; as with all things in deep learning, **we can not rigorously prove its effectiveness or optimality. However, it is well-motivated and performs well in practice.**
>
> There exists [past literature that shows success with optimisation in learning trajectory-based subspaces.](https://ieeexplore.ieee.org/document/9782552)
>
>
> Pretraining on a generic dataset with structure-rich and diverse images moves along a trajectory from random weights towards weights suitable for reconstructions from filtered backprojected inputs for a specific operator. We believe this is a reasonable starting point for data-dependent fine-tuning.
>
> Removing degrees of freedom reduces variance in the estimation of the fine-tuned network parameters. Figure 2 shows how too small dimensional subspaces lead to a substantial degradation in performance. However, the increase in performance with subspace dimension plateaus quickly, around dimension ≈1200.
>
> We attribute this to the top-k SVD decomposition finding the directions in parameter space that are important for task-agnostic training — and these also being the important ones for fine-tuning. We do not want to deal with directions in parameter space that barely affect the output. Dealing with projections of the full parameter space imposes a linear cost in the total number of parameters. We can reduce this cost with "sparsification". The intuition here is the same as with top-k SVD but we reduce the size of the basis vector matrix's other axis. As far as dimensionality reduction goes, these choices are very standard (see for instance [these course notes](https://users.cs.utah.edu/~jeffp/teaching/cs5140-S17/cs5140/L15-MatrixSketching.pdf) and [this optimisation paper](https://proceedings.mlr.press/v161/feng21a/feng21a.pdf)).
>
> ---
>
>
> **2. What sort of biases does dimensionality reduction computed on the ellipses dataset introduce?**
>
> > it seems that the basis vectors used for the dimensionality reduction were extracted from a pretraining scheme based on an ellipse dataset. What sort of biases does this dataset introduce? Does this favor better reconstruction of piecewise smooth images rather than images with fine details? A discussion of this will also strengthen the paper.
>
>
> This is a very good point!
> The ellipses dataset is decently diverse; as it contains many overlapping ellipses of different sizes and orientations. This also creates some sharp discontinuities. In terms of biases, early fine-tuning checkpoints hallucinate some ellipses, but this behaviour disappears with more steps.
>
> **We have compared ellipses pre-training to alternative basis choices**. Comparison with a **randomly selected subspace** is provided in Figure 16 in the Appendix. Randomly selected subspace always performs at least 0.5dB worse than pre-training trajectory sampling. We also considered **pre-training on a task-specific dataset** of images similar to the ones the model would be evaluated on. This leads to better performance, as shown in Figure 21. However, the generality of the approach is lost, and the method starts to resemble supervised learning. Moreover, a task and modality specific dataset of images might not always be available, whereas a synthetically generated one is. Thus, we think that the **ellipses-based pre-training trajectory basis represents a reasonable compromise**.
>
>
> **Our updated draft discusses this in the main text (Section 5.3), making references to the full study in the Appendix A.3.**

---

> > ### Author Response · Authors · 2023-12-20
> > **Response to reviewer aq9Y – Part II**
> >
> > **3. The evaluation methodology — limitations of PSNR**
> >
> > > ...evaluation methodology used is fairly limited .... For example, in the case of medical imaging, improved performance of relevant downstream tasks (eg. detection tasks relevant for diagnosis) are far more important than PSNR. For applications where perceptual quality is the most important PSNR is still not always a good metric since it favors images with oversmoothed features. Thus, the authors should consider using a a better variety of metrics.
> >
> > > Using a wider variety of evaluation metrics as well as task-based evaluation metrics will definitely strengthen the paper.
> >
> >
> > We have **added tables with SSIM values for all natural images** (Set 5). SSIM was designed to better allign with perceptual quality than PSNR, although as the reviewer is surely aware, it is no silver bullet.
> >
> >
> > We fully agree with the reviewer about the limitations of PSNR — it does not reflect perceptual quality and it is not alligned with downstream tasks.
> >
> > Despite its flaws, **PSNR is standard in both industry** ([AVIF, the new image format from Alliance for Open Media (AOM), is tuned for PSNR](https://web.dev/articles/compress-images-avif), [Adobe also uses PSNR to evaluate image formats](https://blog.developer.adobe.com/image-optimisation-with-next-gen-image-formats-webp-and-avif-248c75afacc4)) **and academia** ([1](https://arxiv.org/abs/1711.10925) [2](https://arxiv.org/pdf/2303.05754.pdf), [3](https://arxiv.org/pdf/1707.06474.pdf), [4](https://arxiv.org/pdf/2004.06688.pdf)). It is used in practically all previous work on the DIP. Perceptual and **task-aligned image evaluation is an open problem** and the only truly reliable approach is use of external raters with domain expertise. This is **beyond the scope of this work**, especially considering we apply our method to a range of tasks: denoising, deblurring and CT reconstruction — task specific evaluation for each individual task would be impractical.
> >
> > **We have added a discussion on the limitations of PSNR and SSIM in Appendix A.4**
> >
> >
> > ---
> >
> > **4. Presenting this paper as a preliminary investigation**
> >
> > > I think it would be important to present this paper as a preliminary investigation of a new regularization scheme for DIP, and tone down statements that indicate that this paper brings DIP closer to real-life applications, since based on the results presented, it is hard to conclude this. It is also essential to mention the limitations of the evaluation methods used in the context of real-life applications.
> >
> >
> > We think the reviewer is right. We did not mean to imply that subspace-training should be used as the new default approach for inverse problems in imaging. Instead, we highlight that it may be something practitioners want to consider when engineering their systems. We are also well-aware of, and introduced discussions of, the limitations of PSNR-evaluation (see response 3. above).
> >
> > We have now **clarified this in the introduction and conclusion** and we have removed or toned down the sentences such as: "We aim for this approach to bring DIP-based CT reconstruction methods closer to co-located deployment in real-life settings.".
> >
> > Note that we do run our method on real-world non-simulated datasets like the Walnut.

---

### Review · Reviewer_FJzL · 2023-12-07

**Summary Of Contributions:**

This paper proposes a new method to mitigate overfitting in DIP. It identifies a sparse and low-dimensional subspace for the network parameters and optimizes within this subspace. The sparse subspace is computed from pretraining trajectory, and second order methods are used for optimization thanks to the reduced dimensionality. Experiments show that the Sub-DIP method is sigfinicantly less prone to overfitting, compared to DIP and E-DIP.

**Audience:**

Yes

**Broader Impact Concerns:**

None.

**Claims And Evidence:**

Yes

**Requested Changes:**

1. The authors may want to justify the choice of using pretraining checkpoints to identify the subspace. Why is optimizing within this subspace a good choice? It seems to the reviewer that, a desirable subspace should capture "good" network parameters, for example, the parameters that achieve good reconstruction performance. However, pretraining checkpoints may not provide this.
For example, early checkpoints capture under-trained networks. On the other hand, later checkpoints may not be diverse enough, as the training may have converged. The reviewer is curious about the rationale behind this choice.

2. What is the stopping criterion for DIP, E-DIP, sub-DIP for each dataset? In muCT-walnut experiment, the authors claimed that DIP and E-DIP cannot use the same criterion as sub-DIP. Why is this the case? More information should be provided since "conv." PSNR is largely dependent on the choice of stopping criterion. How did authors select the criterion for each method? Additionally, how is lambda in (2) chosen?

3. Can the authors comment on why using time, instead of iterations, as x-axis? Time is less reliable since it can be affected by many factors such as CPU/GPU load. How would the results look if plotted against iterations?

4. The authors can comment on the requirement of a pretraining dataset for sub-DIP (and E-DIP), and contrast it with the original DIP which does not require pretraining. Furthermore, given access to data samples, other alternatives, such as Plug-and-Play and diffusion models, can be applied. The authors may want to address the pros and cons of these methods, to place their a work in a larger scope.

5. The L_gamma in (6) is different from the one defined in (5) (the former does not include TV). The authors may want to clarify this before deriving $\nabla_c L_\gamma (c)$ to avoid confusion.

6. The reviewer is curious how pretraining can be amortised across different reconstructions, as stated in Step 1.

7. In Figure 1, is "last" the same as "conv."?

8. Figure 2: Is the DIP conv. result missing?

9. "Fig. 2 (right) shows that Sub-DIP LBFGS and NGD converge in less than 50 seconds. These methods are optimal along the known Paretofrontier, with the former reaching ∼2 dB higher reconstruction fidelity and the latter converging faster (in ∼20 seconds)."
It seems LBFGS converges faster and the gap is not as large as 2dB.

10. Consider moving Algorithm 1 to the main body.


Overall this paper presents an interesting novel methodology with promising results. The authors can further improve their work by addressing these points.

**Strengths And Weaknesses:**

**Strengths**

The method is well-motivated, and innovative in integrating techniques from different domains that include subspace estimation, optimization and DIP.

The experiment design, from ablation study to comparison with existing methods, is robust and comprehensive, with good experimental details.

Results show strong performance of the proposed method across multiple datasets and tasks.


**Weaknesses**

Some aspects of method design need more justification, e.g. the rationale behind idetifying the subspace from pretraining checkpoints.

Some important experimental details are unclear, especially the stopping criterion.

---

> ### Author Response · Authors · 2023-12-20
> **Response to reviewer FJzL – Part I**
>
> We thank the reviewer for reading our paper and for their insightful comments. Below we address their comments and questions.
>
> ---
>
> **1. Why use the subspace of pretraining checkpoints?**
>
> > The authors may want to justify the choice of using pretraining checkpoints to identify the subspace. Why is optimizing within this subspace a good choice? ... a desirable subspace should capture "good" network parameters, ... pretraining checkpoints may not provide this. For example, early checkpoints capture under-trained networks. On the other hand, later checkpoints may not be diverse enough, as the training may have converged.
>
>
> The reviewer is right that the choice of subspace is important and non-trivial.
>
> First of all, we would like to point to [past literature that shows success with optimisation in learning trajectory subspaces](https://ieeexplore.ieee.org/document/9782552). We **avoid untrained networks by using an initial burn-in period** of 1000 gradient updates. We then collect snapshots at uniform intervals until the last pre-training gradient update. Our **pre-training set contains 32000 images, lowering the chances of overfitting** during pre-training.
>
> **We have studied alternative sampling strategies**. Comparison with a **randomly selected subspace** is provided in Fig. 16 in the Appendix. The results show that randomly selected subspaces always perform at least 0.5dB worse than those via pre-training trajectory sampling. We also considered **pre-training on a task-specific dataset** of images similar to the ones the model would be evaluated on. This leads to better performance, as shown in Fig. 21 in the revision. However, the generality of the approach is lost, and the method starts to resemble supervised learning. Moreover, a suitable dataset of images that is specific enough for a task and modality at hand might not always be available, whereas a synthetically generated one is. Thus, we think that pre-training trajectory sampling represents a reasonable compromise.
>
> **Our updated draft briefly discusses these studies and refers to them in the Section 5.1 for the CartoonSet and Section 5.3 for the Mayo dataset.**
>
> We also tried taking more samples from either early or late stages of the pre-training trajectory, but found these to perform worse.
>
> ---
>
> **2. How are the stopping criteria for DIP, E-DIP and sub-DIP selected for each dataset?**
>
> >What is the stopping criterion for DIP, E-DIP, sub-DIP for each dataset? In muCT-walnut experiment, the authors claimed that DIP and E-DIP cannot use the same criterion as sub-DIP. Why is this the case? More information should be provided since "conv." PSNR is largely dependent on the choice of stopping criterion. How did authors select the criterion for each method? Additionally, how is lambda in (2) chosen?
>
> As a stopping criterion we use Algorithm 1, with a pre-specified patience $\mathfrak{p}$, proportion $\delta$ and metric $g(\cdot)$. Specifically, we use the optimisation loss in equation (2) as the stopping metric  for CartoonSet and Set5 as:
> * For *CartoonSet* and for *Set5* we use Algorithm 1 with  $\delta=0.995$ and $\mathfrak{p}=100$.
>
> The details for *Set5* were missing from the paper but have now been added to Section 5.4.
>
> For $\mu$CT and *Mayo Clinic* the training loss for DIP and E-DIP behaves very poorly due to the severe overfitting. In the right panel of Fig. 17 we see that the loss is continuingly  (but slowly) decreasing, whereas the top row of Fig. 6 shows that the PSNR peaks early and we then overfit to noise. Instead, motivated by [recent work](https://arxiv.org/pdf/2112.06074.pdf), we attempt to use the moving variance estimator defined as
>
> $ \text{Var}(k) = \frac{1}{M}\sum_{m=0}^M \|x^{k+m}-\frac{1}{M}\sum_{i=0}^{M-1}x^{k+i} \|_F^2,$
>
> where $x^i$ is the image reconstruction at iteration $i$. Moreover,
> * For *$\mu$CT Walnut* dataset we then use Algorithm 1 with $\delta=1$ and $\mathfrak{p}=1000$ for DIP and E-DIP and $\delta=0.995$ and $\mathfrak{p}=100$ for sub-DIP methods.
>
> However, the $\mu$CT results in Fig. 6 show that even this stopping criteria is not a reliable estimator of a good reconstruction quality for DIP and E-DIP. This is discussed in Section 5.2 on page 8.
>
>
> Parameter $\lambda$ is chosen on the validation set, which depends on the given dataset and measurement modality. **We report the details in section D.4 in the Appendix; which are tabulated in Table 1.**

---

> > ### Author Response · Authors · 2023-12-20
> > **Response to reviewer FJzL – Part II**
> >
> > **3. Using time, instead of iterations, as x-axis**
> > > ... it can be affected by many factors such as CPU/GPU load. How would the results look if plotted against iterations?
> >
> > This is an important question.
> >
> > We report wall-clock time because **different optimisation algorithms take very different amounts of compute per-step**. NGD requires solving a linear system, with cubic cost, in each step of the iteration, while the cost of SGD is linear in the number of parameters. **However, NGD makes significantly more progress in each iteration**.
> >
> > We run all methods on the same A100 GPUs. We run the same script for all methods, with only the optimisation code changing. Nothing else is running on the compute server, which is a dedicated HPC node.
> >
> > The **results with respect to the number of optimisation steps are reported in the Appendix**, see Figs. 19-21 for $\mu$CT Walnut, Mayo with 100, and Mayo with 300 angles, respectively.
> >
> > Sub-DIP methods are also somewhat faster in terms of the number of iterations. For example, $\mu$CT Walnut data (Fig 16), SubDIP-NGD reaches its _max_ PSNR in ~100 iterations, whereas DIP takes ~1000. However, due to the additional computational overhead, the required computational times to reach the _max_ values are almost the same between the two methods. For this reason, we felt the optimisation time to be a more relevant and fair metric when comparing the performance.
> >
> > We now explicitly reference the plots showing the performance in terms of iterations in Section 5.3 of the main body of the paper.
> >
> > ---
> >
> > **4. Placing the pros and cons of the work within the larger scope of methods that use data samples (such as PnP, diffusion, original DIP)**
> > > The authors can comment on the requirement of a pretraining dataset for sub-DIP (and E-DIP), and contrast it with the original DIP which does not require pretraining. Furthermore, given access to data samples, other alternatives, such as Plug-and-Play and diffusion models, can be applied. The authors may want to address the pros and cons of these methods, to place their a work in a larger scope.
> >
> > Thanks for raising these comparisons!
> >
> > **Pre-training** the DIP weights to reduce the reconstruction time and computational costs was one of the motivating paradigms of the E-DIP paper, and are discussed therein. However, pretraining also increases the susceptibility to overfitting. We discuss this issue in Section 3, and confirm it in our experiments (see for example the optimisation curves for DIP and E-DIP in Fig. 4). **Sub-DIP reduces the susceptibility to overfitting** via dimensionality reduction in the parameter space.
> >
> > Same as DIP based methods, **Plug-and-Play** algorithms may be trained in an unsupervised manner, not requiring data from the sought image distribution. However, the network primarily operates as a denoising engine.
> >
> > **Diffusion methods** are an example of generative modelling (to which DIP can also be claimed to belong) and they have seen some use in inverse problems. However, this line of research is still in the early stages and as [a work of Wang et al.](https://arxiv.org/pdf/2112.06074.pdf) suggests, their reconstruction performance can be highly sensitive to a distribution mismatch (e.g. in terms of the level and type of noise) in training and testing. The fully unsupervised DIP does not suffer from the same issue.
> >
> > **We now note at the end of section 4 of the main text that our methodology can be extended beyond the DIP framework.**
> >
> > ---
> >
> > **5. $L_\gamma$ used in (6) is different from the one in (5) due to lack of TV. This needs clarification before $\nabla_c L_\gamma(c)$ is derived**
> >
> >
> > This is an important point. Right after equation (8) we state **"TV’s contribution is omitted since the FIM is defined only for terms that depend on the observations and not for the regulariser (Ly et al., 2017)."** Although it is in the same paragraph, there is a pagebreak between equation (6) and this statement. **To avoid confusion, we have moved the statement to right before equation (6).**
> >
> >
> > ---
> >
> > **6. How is pretraining amortised across different reconstructions**
> >
> > This is also an important point.
> > We pre-train a DIP model with the ellipse dataset to generate a subspace for fine-tuning. Once this is done, **we use the same subspace to fine-tune the DIP on all images of interest**. Thus, the pre-training is "amortised" in the sense of paying a cost once but reaping the reward many times.
> >
> > **This is discussed in Step 1 of Section 4.**

---

> > > ### Author Response · Authors · 2023-12-20
> > > **Response to reviewer FJzL – Part III**
> > >
> > > **7. Is "last" same as "conv."?**
> > >
> > >
> > > By _conv._ (convergence) we refer to the reconstruction computed when using the stopping rule provided by Algorithm 1, as described in the initial paragraph of Section 5.1 and in the caption of Fig. 2. On the other hand, _last_ refers to the last available iterate.
> > >
> > > We report _conv._ only where Algorithm 1 is aplicable; for very noisy images like in Fig. 6, the stopping method performs worse than running for a very large number of iterations. In these settings, we report _last_.
> > >
> > > ---
> > >
> > > **8. Is the DIP conv. result missing in Figure 2?**
> > >
> > > Good observation!
> > >
> > > DIP conv. is not missing in Fig. 2, however **it is hard to see as it is almost completely obscured by the green E-DIP conv.** We have **updated the caption** to clarify this point.
> > >
> > > ---
> > >
> > > **9. Sub-DIP LBFGS seems to converge faster than Sub-DIP NGD, and the gap is not 2dB**
> > > > "Fig. 2 (right) shows that Sub-DIP LBFGS and NGD converge in less than 50 seconds. These methods are optimal along the known Paretofrontier, with the former reaching ∼2 dB higher reconstruction fidelity and the latter converging faster (in ∼20 seconds)." It seems LBFGS converges faster and the gap is not as large as 2dB.
> > >
> > > We thank the reviewer for pointing this out! The phrasing in the original manuscript was confusing.
> > >
> > > **We were referring to the fact that Sub-DIP NGD and LBFGS both had 1.5-2dB better performance than E-DIP and regular DIP**, and that they also converge faster with Sub-DIP LBFGS requiring only 20 seconds and Sub-DIP NGD requiring 50 seconds to converge.
> > >
> > >
> > > **The corresponding text has been updated to**: "Fig. 2 (right) shows that Sub-DIP L-BFGS and NGD converge in less than $50$ seconds. These methods are optimal along the known Pareto-frontier, reaching $\sim1.5$ dB higher reconstruction fidelity than DIP and E-DIP. LBFGS converges in only $\sim 20$ seconds."
> > >
> > > ---
> > >
> > > **10. Consider moving Algorithm 1 to the main body.**
> > >
> > >
> > > Algorithm 1 is now in the main body of the paper.

---

> > > > ### Comment · Reviewer_FJzL · 2024-01-04
> > > >
> > > > Thank the authors for clearing all my confusions and questions.
> > > >
> > > > I just want to note a few points on the response:
> > > > 1. In the equation for Var(k), \ell should be m.
> > > > 2. Some figure indices are from the new manuscript, while some are from the original, without clear definitions. This caused a bit problem in readability.
> > > >
> > > > I have no additional questions or comments.

---

### Author Response · Authors · 2023-12-20
**Thank you for your comments**

We thank all reviewers for their time reading our work and their comments.

We have now incorporated the feedback into an updated draft. We would like to point out that we added two additional experimental studies suggested by the reviewers:

1. the interplay between noise in the data to be reconstructed and the dimension of the extracted subspace
2. the sensibility of the extracted subspace to shift in the forward operator.

---

### Decision · Action_Editor_TdNh · 2024-01-21

**Recommendation:** Accept as is

**Comment:**

The ideas proposed in this paper are novel and intuitive. While there is no theoretical analysis that supports the methodology, extensive empirical results support the authors claims and provide an advancement with respect to state of the art in image restoration with Deep Image Prior. All reviewers support the acceptance of this paper, and I'm recommending this accordingly.

In preparing the camera ready version of this paper, I encourage the authors to address the reviewers comments that haven't been addressed already, if any, as well as revise for typos and inaccurate statements (e.g., the expression $d_\text{pre} = \mathcal O (10^3)$ does not make sense from an O-notation perspective, since $10^3$ is a constant; I presume the authors simply mean "in the order of", or $d_\text{pre} \approx 10^3$ ).

**Audience:**

This paper will be of interest to people working on deep-learning based models for inverse problems and unsupervised learning.

**Claims And Evidence:**

This paper proposes ways to alleviate the problem of overfitting and large training times in employing the Deep Image Prior approach for unsupervised image restoration and reconstruction. The authors claim that better behaviour (less overfitting) and efficiency is possible if the model weights are restricted to a sparse linear subspace. All reviewers agree that these claims are well supported by comprehensive experiments, and value the contribution.

---

> ### Author Response · Authors · 2024-02-18
> **Thank you**
>
> We have now uploaded the camera-ready version of our paper with the requested changes and additional experimental investigations suggested during the review process.
>
> Best regards,
> The Authors